# A Semantic Partition Algorithm Based on Improved K-Means Clustering for Large-Scale Indoor Areas

**Kegong Shi, Jinjin Yan \*** and **Jinquan Yang**

Qingdao Innovation and Development Center, Harbin Engineering University, Qingdao 266500, China;
kegong.shi@hrbeu.edu.cn (K.S.); yjq980314@hrbeu.edu.cn (J.Y.)
\* Correspondence: jinjin.yan@hrbeu.edu.cn

**Abstract:** Reasonable semantic partition of indoor areas can improve space utilization, optimize property management, and enhance safety and convenience. Existing algorithms for such partitions have drawbacks, such as the inability to consider semantics, slow convergence, and sensitivity to outliers. These limitations make it difficult to have partition schemes that can match the real-world observations. To obtain proper partitions, this paper proposes an improved K-means clustering algorithm (IK-means), which differs from traditional K-means in three respects, including the distance measurement method, iterations, and stop conditions of iteration. The first aspect considers the semantics of the spaces, thereby enhancing the rationality of the space partition. The last two increase the convergence speed. The proposed algorithm is validated in a large-scale indoor scene, and the results show that it has outperformance in both accuracy and efficiency. The proposed IK-means algorithm offers a promising solution to overcome existing limitations and advance the effectiveness of indoor space partitioning algorithms. This research has significant implications for the semantic area partition of large-scale and complex indoor areas, such as shopping malls and hospitals.

**Keywords:** area semantic partition; improved K-means; large-scale indoor areas

## 1. Introduction

With the rapid urbanization and expansion of indoor environments in large-scale settings [1,2], the partition of indoor areas has become a significant concern [3]. Reasonable partitioning plays a crucial role in improving space utilization, optimizing management, and enhancing safety and convenience. It has wide-ranging applications in various fields such as indoor navigation [4,5], security monitoring [6], and resource management [7,8]. A semantic-based area partition is able to bring in numerous benefits to large-scale indoor scenes (such as shopping malls and hospitals), including improved business operations, enhanced customer service, and increased safety, leading to better experiences. For instance, in shopping malls, a semantic-based area partition can assist new businesses in selecting suitable shop locations based on attributes of their products. This allows the mall to effectively manage the procurement, sales, and inventory of different product categories, thereby enhancing overall operational efficiency. In particular, such a partition can assist shopping malls in determining suitable locations for signage installation, which can effectively help customers quickly locate the desired products and save shopping time. Analyzing rest areas enables the placement of flammable and explosive items in safe zones away from those areas, mitigating potential hazards and safety risks. In hospitals, dividing areas based on different departments (semantic categories) improves the efficiency of patient diagnosis and treatment. Patients can more accurately determine the areas they need to visit, like inpatient wards and diagnostic sections, which helps in reducing wait times and confusion, thereby improving the overall patient experience.

Indoor area partition can be seen as an application of regional clustering [9]. K-means is a widely utilized machine learning algorithm among various partitioning methods [10].

It is used to divide *n* observations into *k* clusters. In this algorithm, each observation is allocated to the cluster whose mean is closest, with this mean acting as the prototype for the cluster [11]. The K-means algorithm can effectively partition indoor areas by specifying the desired number of clusters, but it encounters various challenges when applied to real-world scenarios. These include its inability to incorporate semantic information about the indoor space, slow convergence, susceptibility to initial clustering center selection, and sensitivity to outliers. To effectively partition large-scale indoor areas, it is crucial to develop an algorithm that takes into account semantic information, achieves faster convergence, and minimizes the impact of initial clustering center selection. The primary consideration lies in the fact that K-means offers the benefits of simplicity and straightforward implementation. Moreover, it facilitates the direct determination of the number of partitions by manipulating the *K*. This is very suitable for the indoor partitioning problem we need to solve. This paper proposes an improved K-means clustering algorithm (IK-means), which aims to partition large-scale indoor areas based on semantics to improve the accuracy and efficiency of large-scale indoor space partitioning. The IK-means employs a blend of Value Difference Metric (VDM) distance and Euclidean distance to compute the distance between the mean vector and samples. In each iteration, the algorithm updates the mean vector to expedite the convergence rate. In order to verify the accuracy and efficiency of IK-means, we conducted a comparison test. Among them, traditional K-means and and density-based spatial clustering of applications with noise (DBSCAN) algorithms [12] were chosen as comparisons, and the two methods were implemented in a large-scale indoor scene.

The presented IK-means exhibits several advantages over the conventional algorithm.

1.  Our algorithm, distinct from traditional K-means and DBSCAN algorithms, incorporates semantic information from known indoor room node attributes. This approach leads to partitioning results that exhibit lower values in evaluation functions. Most notably, it eliminates the need to grapple with a multitude of redundant and intricate variables, enabling swift and automated generation of partitioning outcomes. This not only leads to cost savings but also yields partition results that closely align with manual partitioning results.
2.  In comparison to K-means and DBSCAN algorithms, our algorithm exhibits faster convergence rates. Experimental results demonstrate that our algorithm, IK-means, achieves a remarkable 93.85% faster convergence than traditional K-means and an impressive 83.29% faster convergence than DBSCAN.
3.  The IK-means algorithm offers a more streamlined process for parameterization. Traditionally, selecting the optimal value for *K* in K-means clustering has been a significant challenge. In contrast, the IK-means approach predefines the parameter *K* based on the semantic information of the rooms, thereby obviating the need to assess the impact of *K* on the partition results. This method effectively addresses and mitigates the issue of parameter dependence that is prevalent in the standard K-means algorithm and its variants.

The remainder of this paper is organized as follows. The next section introduces and compares seven algorithms commonly employed for regional clustering. Section 3 presents the specific details of IK-means. In Section 4, we demonstrate the feasibility of IK-means by a series of experiments in an indoor area. Lastly, Section 5 provides a comprehensive summary of the entire text and discusses potential future research directions.

## 2. Related Work

Indoor area partition can be seen as an application of regional clustering [13]. The commonly used algorithms for clustering include K-means [14] and its variants, DBSCAN [12], the hierarchical clustering algorithm [15], the spectral clustering algorithm [16], and other cluster algorithms [17–19].

K-means is an algorithm often used for data clustering, which divides a dataset into *K* different clusters based on the similarity of the data. The algorithm updates the partitioning outcome by iteratively computing the distance between the nodes and cluster

centers. The primary advantage of K-means is its simplicity and ease of implementation. Nonetheless, the algorithm necessitates a predetermined number of clusters $K$, and the initial cluster centers have an impact on the final clustering results. Thus, several improved versions of K-means were developed, including K-medoids [20], Kernel K-means [21], and K-means++ [22].

The K-medoids [20] is a clustering algorithm that uses cluster centers, known as medoids, to represent their respective clusters. Unlike the traditional K-means algorithm, which calculates the distance between each data point and every cluster center in each iteration, the K-medoids utilizes medoid selection based on the lowest average distance to other nodes and assigns them as cluster centers. However, the selection of medoids requires calculating the cost, which can be computationally intensive. Consequently, when applied to large-scale datasets, K-medoids tends to exhibit a slower performance.

Kernel K-means [21] addresses the challenge of linearly indistinguishable clusters in the original data space by mapping the data to a higher-dimensional space. It utilizes polynomial kernel functions, Gaussian kernel functions, and other techniques to enhance its clustering model. A notable strength of Kernel K-means lies in its ability to process complex, particularly nonlinear, datasets effectively, without necessitating prior knowledge of the cluster count $K$. Additionally, the algorithm offers better interpretability compared to some other clustering methods. However, the computational efficiency of Kernel K-means may be compromised due to the high computational complexity associated with the kernel function. The performance of the Kernel K-means algorithm is also heavily reliant on the choice of the kernel function and its parameters, highlighting the criticality of selecting suitable kernel functions and parameter configurations to achieve meaningful clustering results.

The K-means++ algorithm [22] solves the problem of local optima that can occur when initial clustering centers are randomly selected by utilizing a specific probability distribution for center selection. This algorithm ensures that the distances between different clustering centers are relatively large, reducing the risk of the algorithm converging to local optima and improving both the accuracy and stability of the clustering results. Therefore, the K-means++ is mainly employed for selecting initial clustering centers to enhance the stability and quality of the algorithm.

DBSCAN [23] is a prominent clustering algorithm that identifies clusters based on high-density regions within a dataset, while treating low-density areas as noise or border points. One major advantage of DBSCAN compared to traditional clustering algorithms like K-means is its ability to automatically determine the number of clusters and identify clusters with arbitrary shapes. However, to achieve optimal performance on different datasets, this algorithm still requires manual parameter tuning, specifically for the radius parameter ($r$) and the density threshold ('MinPts'). Furthermore, when dealing with high-dimensional data, DBSCAN may encounter the challenge of the "curse of dimensionality", which can lead to decreased clustering performance. Nonetheless, DBSCAN remains a valuable tool for clustering tasks, particularly in scenarios where the number of clusters is unknown and the data exhibit complex structures.

The hierarchical clustering algorithm [24] clusters datasets by iteratively merging data points to form a hierarchical structure. One of the main advantages of this algorithm is its ability to generate a clustering hierarchy, allowing for better visualization and analysis of the results. The method is versatile and can be applied to various datasets and clustering tasks as it does not require a predetermined number of clusters. However, the computational complexity of hierarchical clustering is high, and it can be time-consuming, especially when dealing with large datasets. Moreover, it is sensitive to the initial clustering configuration, often requiring multiple experiments to achieve optimal clustering results. Despite these challenges, it remains a valuable algorithm for exploring and understanding the inherent structure of datasets.

The spectral clustering algorithm [25] is a graph-based method that treats data points as nodes in a graph and constructs the graph by measuring pairwise similarity between

nodes. It then transforms the problem of spectral decomposition of the graph into a feature vector problem, and clustering is performed on these feature vectors. This algorithm shows several advantages, including its ability to handle nonlinear clustering problems and its applicability to large-scale datasets. Unlike traditional distance-based clustering methods, spectral clustering does not require a predefined number of clusters, providing flexibility in the clustering process. However, selecting an appropriate similarity measure when constructing the similarity matrix is crucial for the success of spectral clustering. Moreover, this method is more sensitive to noisy data and outliers compared to some other clustering algorithms. Despite these considerations, it is a valuable algorithm for tackling complex clustering tasks and has found applications in various domains.

However, the aforementioned algorithms are not well-suited for indoor area partitioning due to their limitations in considering semantic information, dependence on parameter choices, and relatively slow convergence rates. Therefore, there is a need for a novel algorithm that effectively addresses these shortcomings. The ideal algorithm should be efficient, capable of incorporating semantic information about indoor spaces, able to handle high-dimensional data, and robust against noisy data and outliers. After evaluating the strengths and weaknesses of existing algorithms, this study has chosen to improve the traditional K-means clustering algorithm. As a result, a novel algorithm called the Improved K-means algorithm (IK-means) has been developed.

## 3. The Improved K-Means Clustering Algorithm for Semantic-Based Partition

In this section, we present the IK-means tailored for semantic-based partitioning. This algorithm leverages the VDM to measure the distance between nominal attributes and subsequently normalizes this distance following its amalgamation with the Euclidean distance. The proposed distance calculation approach enables the comprehensive integration of semantic information pertaining to indoor area attributes and physical distances of spatial data, ultimately bringing in a more rational partitioning. The algorithm encompasses four critical stages, including the determination of $K$, the clustering of common nodes, the fixing of orphan nodes, and the reconstruction of clusters.

### 3.1. Concepts and Parameters

We used a node-relationship graph (NRG) model to model the indoor environment map. NRG can abstract indoor regions as nodes and represent the connectivity between regions through edges, and it provides an analytical basis for the subsequent determination of the rationality of partitioning. Therefore, in this paper, we employ the NRG model to establish an indoor map model. The theoretical underpinning for generating NRG from indoor maps is Poincaré duality [26].

The indoor spaces comprise rooms, corridors, staircases, and doors. The first three are geometrically represented as polygons, while doors are depicted as lines. To achieve NRG, room and stair spaces are further reduced to nodes, specifically using the centroids of their respective polygons. In the case of doors, the mid-points of the lines are employed as their representative nodes. When two indoor spaces are interconnected by a door or doors, two edges are established to link the mid-point of the door lines with the centroids of the respective spaces. For corridor spaces, we utilize their mid-line to connect with the mid-plumb of the door lines. In particular, the door nodes are projected onto the mid-line of the corridor, and the resulting vertical point serves as the corridor node. Figure 1a depicts an indoor map, while its corresponding NRG is presented in Figure 1b.

In the generated NRG model, we introduce three fundamental concepts to characterize both the edges and nodes.

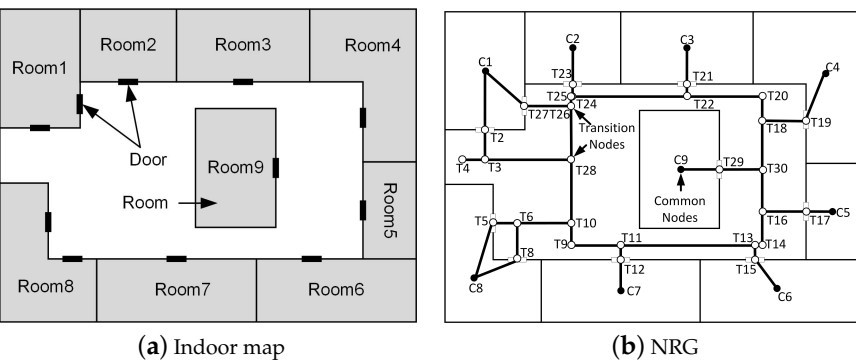

**(a)** Indoor map        **(b)** NRG

**Figure 1.** Example of a map of an indoor scenario and its corresponding NRG.

- Common nodes ($C_i$): These nodes have specific functional attributes, such as room, elevator/stairs/escalator nodes. Common nodes are characterized by attributes, including identification numbers, semantic information, geometric coordinates, and connected edges. The semantics of common nodes come from the usage of the spaces that they represent. For instance, in a hospital, these usages of spaces may include wards, consultation rooms, and functional areas. During the process of IK-means, common nodes serve as the sample set and participate in the entire process.
- Transition nodes ($T_i$): Unlike common nodes, these nodes are intermediate nodes that serve to connect common nodes, representing locations without specific meaning. Examples of transition nodes include door nodes and corridor nodes. Transition nodes typically lack special attributes but possess attributes like identification numbers, geometric coordinates, and connected edges.
- Edges: Edges typically represent the connections between room nodes, door nodes, and corridor nodes. They are characterized by attributes such as identification numbers and information about the nodes they connect. The presence of edges is crucial in determining the existence of any orphans or disconnections within the resulting clusters after partitioning.

Other than that, the primary parameters governing the IK-means include the following:

- Number of clusters ($K$): This parameter signifies the desired number of clusters and plays a pivotal role in determining the clustering quality, model complexity, and computational cost [27]. The appropriate choice of $K$ is crucial for the K-means [28], as it directly impacts the results.
- Maximum number of iterations ($m$): This parameter sets the upper limit for the number of iterations the algorithm will undergo. It is an important factor in controlling convergence and algorithm termination.
- Control parameters ($W$): These parameters govern the influence of nominal attributes and the physical distance metric. They allow the fine-tuning of attribute importance within the clustering process.
- Number of nearest neighbor nodes ($o$): This parameter determines the number of nearest neighbor nodes considered during cluster transition node grouping.
- Minimum number of nodes in a cluster ($p$): This parameter sets the minimum threshold for the number of nodes within a cluster. Clusters with fewer than this number of nodes are not retained during the cluster reconstruction phase.

### 3.2. Determination of K

The process of selecting initial clustering centers usually involves a certain level of randomness. In traditional K-means, initial clustering centers are typically chosen at random from the dataset, with $K$ node samples serving as these starting nodes. The random selection method can significantly influence clustering outcomes, often resulting in substantial variability in classification results and less than optimal partitioning effects [29].

In the IK-means algorithm, a more systematic approach is adopted to determine the number of partitions (*K*). In particular, it takes into account the semantic information of clustering samples during the initialization of clustering centers. This means that, prior to the clustering process, the number of partitions is determined based on the quantity of distinct semantic features present in the data. Subsequently, the process involves randomly selecting one node from the group of nodes that exhibit unique semantic features, to serve as an initial clustering center. This procedure is repeated until *K* initial clustering centers are established, with each center representing a unique semantic attribute.

This approach ensures that diverse semantic information is represented in each clustering center. Consequently, the default semantic distances between the cluster centers are relatively large when calculating feature distances, making it easier to classify nodes with similar semantic information into the appropriate clusters.

The adoption of such a systematic approach is driven by the utilization of semantic information to accomplish this. For instance, rooms in different environments, such as shopping malls or hospitals, often exhibit distinct semantic characteristics. These semantic attributes are used to pre-determine the number of semantic categories among all common nodes, effectively establishing the value of *K*. For example, rooms in shopping malls may possess semantic features like consumption areas, functional areas, and office areas [30], while hospitals may display a range of semantic attributes such as inpatient departments, outpatient departments, and functional areas [31]. This semantic-driven approach not only ensures a suitable value for *K* but also minimizes variations in partition results arising from different *K*. It is important to note that the configuration of the lower limit for nodes in the transition node and reconstructed cluster, which is often influenced by the map size, does not have a direct impact on the clustering outcome. The significance and influence of these parameters will be further elaborated upon during the introduction of the algorithm.

Therefore, in the context of large-scale indoor environmental partitioning, this method of selecting clustering centers enhances classification accuracy and mitigates disparities in partitioning results caused by different choices of cluster centers. This addresses the shortcomings of the traditional K-means algorithm.

### 3.3. Clustering of Common Nodes

After determining the *K*, we initiate the clustering of common nodes. These common nodes that require classification comprise the node example set *D*. The formation of clusters *K* is contingent upon the primary semantic features of these common nodes. These primary semantic features represent the classifications of semantic information for these nodes and serve as the basis for partitioning. Establishing the required number of partitions *K* based on the semantic feature types of common nodes in the node example set *D*. Subsequently, from the common nodes that exhibit *K* distinct types of semantic features, arbitrarily selecting one node from each of the *K* feature types. These selected common nodes, totaling *K*, serve as the initial clustering centers, denoted as $\mu_1, \mu_2, \cdots, \mu_K$.

Following this, we proceed to examine all common nodes in the node example set *D*, calculating their distances to each cluster center. In this process, we employ two distance metrics: $D_{VDM}$ and $D_{Euclidean}$. These metrics are normalized using min-max normalization and then weighted to compute the sum. Specifically, $D_{VDM}$ is used to calculate the distance for nominal attributes (Equation (1)). The VDM distance cannot be calculated if the attributes of the node differ from those of all evaluated nodes in the cluster, resulting in an infinite VDM distance. For such cases, $D_{VDM}$ is directly normalized as 1. Nominal attributes refer to those attributes that do not have a natural order or arrangement. In this paper, this term specifically refers to the semantic information of nodes. The number of nominal attributes is determined by different datasets, and these datasets can calculate the semantic distance of multiple unordered attributes using Equation (1).

Additionally, the coordinates are utilized as physical attributes to compute the physical distance between nodes.

$$D_{VDM}(x_{il}, w_{il}) = \begin{cases} \sqrt{\sum_{c=1}^{q} \left| \frac{N_{l},x_{il},c}{N_{l},x_{il}} - \frac{N_{l},w_{il},c}{N_{l},w_{il}} \right|^2} & , l = 1, 2, \cdots, m_c, \end{cases} \tag{1}$$

where $x_{il}$ is the $l$-th attribute of instance $x_i$. $w_{il}$ is the $l$-th attribute of instance $w_i$. $q$ is a parameter that determines the influence of frequency differences on the distance calculation. $N_l, x_{il}, c$ is the number of times cluster c appears with the attribute value $x_{il}$. $N_l, w_{il}, c$ is the number of times cluster c appears with the attribute value $w_{il}$. $N, x_{il}$ is the total number of instances where the attribute value is $x_{il}$. $N, w_{il}$ is the total number of instances where the attribute value is $w_{il}$. $m_c$ is the total number of clusters in the dataset.

$D_{Euclidean}$ signifies the Euclidean distance (Equation (2)), which is employed to establish the physical distance between nodes.

$$D_{Euclidean}(x_i, x_j) = \sqrt{\sum_{l=1}^{m_n} |x_{iu} - x_{ju}|^2}, \tag{2}$$

where $x_{iu}$ and $x_{ju}$ are the coordinates of $x_i$ and $x_j$, and $m_n$ is the dimension of the coordinates.

The min-max normalization process scales the values of these two distances to a range between 0 and 1, mitigating the unit effect of the two distance metrics (Equation (3)).

$$y_i = \frac{x_i - \min}{\max - \min}. \tag{3}$$

Additionally, we introduce an extra parameter, denoted as $W \in [0, 1]$. This parameter is used to control the ratio between $D_{VDM}$ and $D_{Euclidean}$, with $W$ being relevant to both the nominal attribute distance metric and the physical distance metric. The value of $W$ needs to be determined experimentally in different datasets, where the selection of $W$ can be determined based on the classification objectives and experimental findings.

Ultimately, we derive the heterogeneous value difference metric (HVDM) distance (Equation (4)). With this metric, we can determine the cluster center that is closest to the current node by calculating the distance between the current node and each cluster center. The current node is then assigned to the respective cluster. We also compute the midpoint between the current node and the cluster center using Equation (5), designating it as the new cluster center.

$$D_{HVDM}(x_i, x_j) = \sqrt{W * \sum_{l=1}^{m_n} D_{Euclidean}^2 + (1 - W) * \sum_{l=1}^{m_c} D_{VDM}^2} \tag{4}$$

$$\mu_i' = \frac{1}{|c_i|} \sum_{x \in c_i} x. \tag{5}$$

During the whole process, certain stopping conditions are established to prevent the algorithm from entering an infinite loop. In the case of clustering common nodes, two stopping conditions are defined: (i) the maximum number of iterations is reached. That is, the clustering process concludes when the number of iterations reaches $m$; (ii) none of the clusters of nodes have changed after traversing all nodes in $D$. To meet these stopping conditions, an upper limit threshold for iterations, denoted as $m$, is pre-set. The selection of $m$ is determined by the size of the dataset [3]. The outcome is the partition of the data into clusters, represented as $C = \{C_1, C_2, \cdots, C_k\}$. Algorithm 1 provides a detailed depiction of the steps involved in IK-means clustering. Node example set $D$ represents the collection of nodes involved in clustering, and $x_i$ denotes each individual cluster sample, specifically referring to nodes participating in the clustering process, and m represents the total number

of samples participating in clustering. Using the example shown in Figure 1 to verify the running effect of Algorithm 1, the result of common nodes clustering is shown in Figure 2.

---

**Algorithm 1** IK-means clustering

---

**Input:** node example set $D = \{x_1, x_2, x_3, \cdots, x_m\}$, number of clusters $K$, threshold $m$
**Output:** Cluster partition $C = C_1, C_2, \cdots, C_k$
sample is randomly selected as the initial clustering centers $\{\mu_1, \mu_2, \cdots, \mu_k\}$ from each of the corresponding attribute categories
Let $C_i = \Phi(1 \le i \le k)$
**repeat**
  **for** $j = 1, 2, \cdots, m$ **do**
    Calculate the distance between $x_j$ and each clustering center $\mu_i(1 \le i \le k)$: $Dist = D_{HVDM}(x_j, \mu_i)$
    Determine the cluster labeling of $x_j$ based on the nearest clustering center: $\Lambda_j = argmin\{d_{ji\,i\in 1,2,\cdots,k}\}$;
    assigning the sample $x_j$ to the corresponding cluster: $C_{\lambda j} = C_{\lambda j} \bigcup \{x_j\}$;
    Compute the new clustering center for the clusters added to the sample: $\mu'_i = \frac{1}{|C_i|}\sum_{x\in C_i} x$
    **if** $\mu'_i \ne \mu_i$ **then**
      Update $\mu_i$ to $\mu'_i$
    **else**
      Keep $\mu_i$
    **end if**
  **end for**
**until** meet the stopping conditions
**return** $C = \{C_1, C_2, \cdots, C_k\}$

---

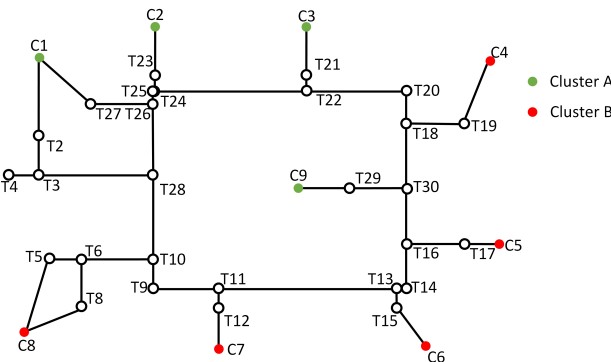

**Figure 2.** Clustering of common nodes.

*3.4. Fixing of Orphan Nodes*

By observing the results of common node clustering, we found that clustering cannot fully meet our requirements, because there are some unreasonable nodes in the clusters. Here, we name the unreasonable nodes orphan nodes. An orphan node is a node that belongs to a different cluster rather than its nearest geometrically associated neighboring nodes, although it is geometrically distant from the nodes in its cluster. This can be explained by the fact that, during clustering, the semantic distance between nodes and the cluster center is considered with priority, and then the physical distance. Moreover, the connectivity among all nodes within the same cluster in the NRG is not taken into account, which frequently leads to the emergence of orphan nodes during the clustering process. However, in real-world observations, when nodes belonging to different semantic categories exist within the same area, the general approach is to include such nodes in the respective partition. Figure 3 shows an example of orphan nodes, in which A, B, and C are three clusters. Nodes in clusters A, B, and C are shown in red, blue, and green, respectively.

This shows that A and C only can be connected indirectly via B. Yet, node a and node b are obviously closer to the nodes in cluster B and the node of B separates it from its slave of the cluster, so *a* and *b* are orphan nodes. In the clustering of Figure 2, orphan nodes are also generated. $C_9$ is a node in cluster A, but its closest nodes in the physical sense are $C_4$ and $C_5$, so $C_9$ is determined to be an orphan node in cluster A. The orphan nodes are circled in blue in Figure 4.

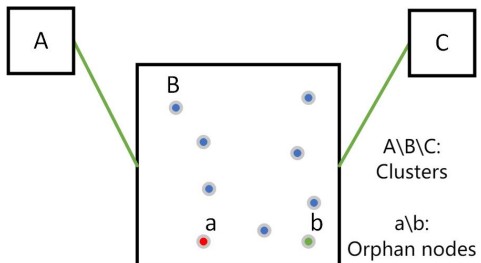

**Figure 3.** Illustration of orphan nodes.

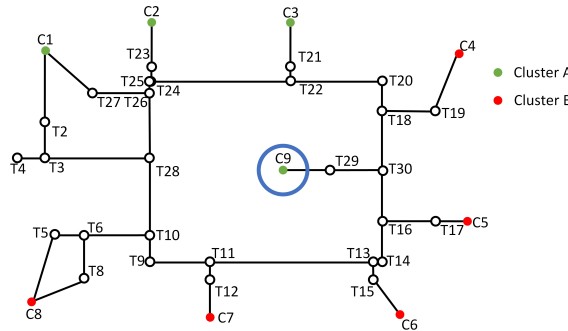

**Figure 4.** Illustration of an orphan node in an NRG.

The presence of orphan nodes may result in numerous transition nodes being erroneously classified into the same cluster as the orphan node. Hence, it is essential to rectify the cluster to which the orphan node belongs. In particular, we calculate the Euclidean distance of each common node from the other common nodes in the sample set *D* and determine the clustering of the *n* nearest neighbor nodes (where the *n* is determined as appropriate). If the number of neighboring nodes belonging to different clusters surpasses the number of clusters to which the node belongs, then the node's cluster is updated to match the cluster hosting the maximum number of neighbor nodes. If the quantity of neighboring nodes in different clusters equals the number in its own cluster, the original cluster remains unchanged. This method effectively resolves the orphan node issue, with orphan nodes from different clusters now reclassified to the cluster that aligns with their surrounding neighbor nodes. Figure 5 shows that, after the correction, the orphan node $C_9$ (in Figure 4) has been changed to cluster B.

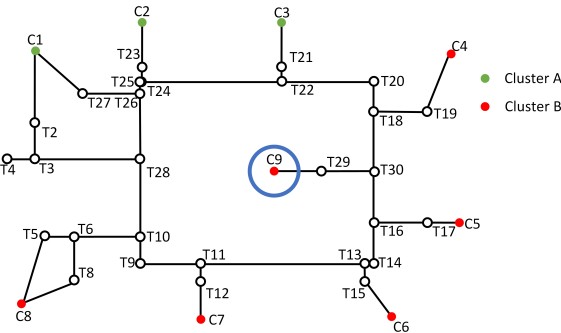

**Figure 5.** Error correction of orphaned nodes.

*3.5. Clustering Transition Nodes and Reconstructing of Clusters*

After repairing orphan nodes, transition nodes will be integrated into clusters. Each common node has already been assigned to a cluster through clustering, and transition nodes will be allocated to clusters based on the cluster attributes of these common nodes. To determine the appropriate cluster for each transition node, the physical distance between the transition node and all common nodes is first computed. Since transition nodes lack semantic information, they are assigned to clusters solely based on physical distance. This is done by calculating and comparing distances using Equation (2), and recording the '*o*' nearest common nodes. Subsequently, within the range of the '*o*' nearest neighboring common nodes of the transition node, the cluster that includes the majority of these common nodes is identified (where '*o*' is defined by the dimensions of the NRG). The transition node is then assigned to the cluster with the most common nodes among these '*o*' nodes. If multiple clusters have an equal and maximal number of common nodes among the '*o*' neighboring nodes, the transition node is randomly assigned to one of these clusters. This process continues until all transition nodes have been added to the clusters.

After clustering the transition nodes into appropriate clusters, node connectivity within each cluster is evaluated using the NRG. This is essential because, during the clustering of common nodes and the addition of transition nodes, connectivity within the map is not considered, which can result in nodes within the same cluster not being directly connected on the map. Therefore, we assess node connectivity within each cluster using the NRG, as all nodes within a cluster should be inter-connected to ensure mutual reachability. In our predefined NRG, edge connectivity represents the linkage between two spaces. If any clusters contain subsets of disconnected nodes, we first check if the number of nodes in these subsets exceeds a predefined threshold parameter $P$. If it does, these subsets are reorganized into new clusters based on their node counts. If the node count in these new clusters falls below the threshold of $p$, the nodes are then merged into another cluster with which they have connectivity. Figure 6 illustrates the disconnected nodes within a cluster. The line signifies the connectivity between clusters. In cluster B, there are two isolated regions. This requires rectification in a subsequent step by splitting and restructuring them into new clusters, or integrating them into existing clusters.

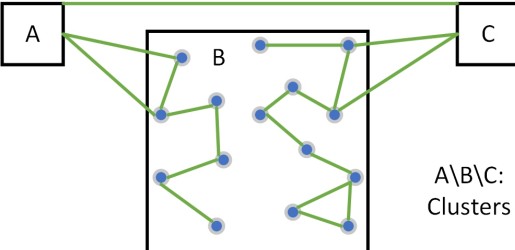

**Figure 6.** Disconnected nodes within a cluster.

Figure 7a illustrates the scenario following the assimilation of transition nodes into the clusters. However, within the blue circle, $C_8$, $T_5$, $T_6$, and $T_8$ in cluster B are isolated by node $T_{10}$, resulting in intra-cluster disconnection. If $p$ is set to 3, $C_8$, $T_5$, $T_6$, and $T_8$ are reconstructed into a new cluster C (Figure 7b). The other cluster B nodes remain unchanged by their own clustering. The light blue nodes represent the reconstruction of the new cluster C. It can be seen that each node has its own cluster information. However, within the blue circle, nodes $C_8$, $T_5$, $T_6$, and $T_8$ are isolated by node $T_{10}$, resulting in intra-cluster disconnection. The issue of disconnected nodes within a cluster refers to the absence of connectivity between nodes within the same classified cluster.

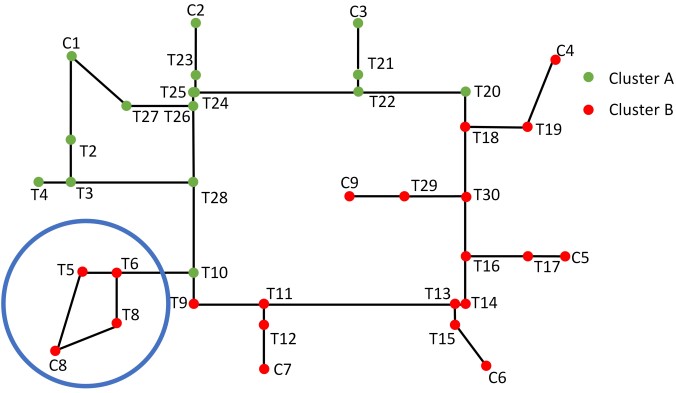

(**a**) Add transition nodes into clusters.

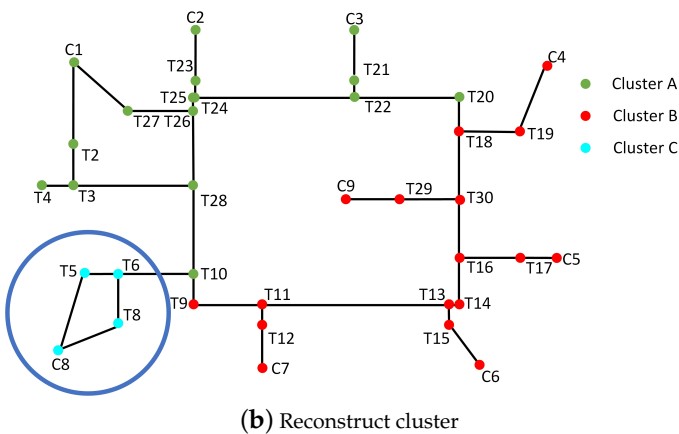

(**b**) Reconstruct cluster

**Figure 7.** Clustering transition nodes and reconstruction of clusters.

## 4. Implementation and Case Study

### 4.1. Case Description and Data Preparation

We modeled an indoor scenario for testing IK-means. The dataset integrates the semantic information of each node and defines the semantically similar regions as the same attribute, that is, the space utilization type, which takes into account the inherent semantic characteristics of each node and facilitates the calculation of its semantic distance. It includes rooms, corridors, elevators, stairs, escalators, atrium, and doors, in which doors are modeled as line segments while all the other elements as polygons (Figure 8a). On the basis of Poincaré duality theory, the scenario is further modeled as an NRG with 496 nodes (Figure 8b). The room's nodes (in purple) are used as common nodes for IK-means, while the corridor's nodes, door nodes, elevator and staircase nodes, and entrance/exit nodes are transition nodes, which are colored in purple, orange, light green, and dark green, respectively. Each node has three attributes: {ID, Type, Attribute}. "ID" records the id of each nodes, "type" keeps the type of nodes and determines whether the node participates in IK-means (e.g., "Common nodes", "Transition nodes"), "attributes" records the specific semantic representations of the nodes (e.g., "conference rooms", "stuff offices", "door nodes", "corridor nodes").

The dataset for the indoor environment consists of a large number of nodes and complex node relationship graphs, making it suitable for our algorithm's research objectives. For larger graphs, the multi-layer structure is typically expanded to address 3D clustering problems. Conversely, smaller graphs lack diversity and are too compact to fully adapt to semantic environments. In such cases, manual partitioning may be more appropriate for dataset selection, ensuring targeted and distinctive characteristics.

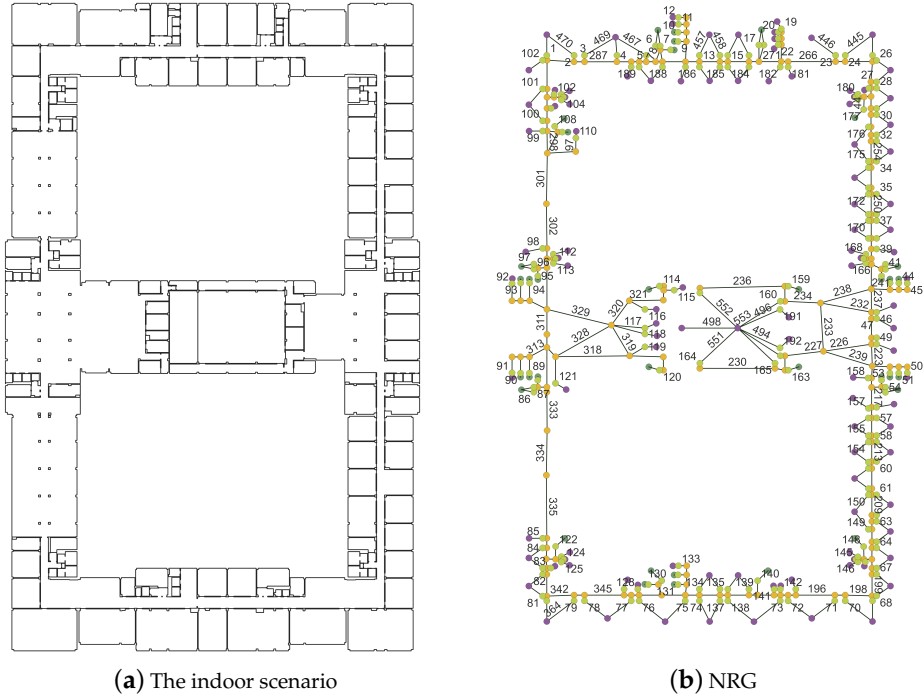

(**a**) The indoor scenario      (**b**) NRG

**Figure 8.** The indoor scenario and its NRG for experiments. The colors of nodes indicator their types, in which the purple, yellow, light green, and dark green are room nodes, corridor nodes, door center nodes, and stair nodes, respectively. Here the room nodes are common nodes. Numbers are IDs of nodes.

In the original dataset, we refer to the real-world observations and employ artificial partitioning to partition the common nodes into *K* regions, maintaining a fixed number of nodes in each region. By quantifying the disparity between the clustering algorithm and real-world observations, we can assess the magnitude of the loss function, thus enabling judgment on the merits and demerits of the partitioning results. The specific loss function is denoted by Equation (6).

$$F = \frac{\sum\limits_{i=1}^{k} \left( \frac{|N_c - N_i|}{N_i} * 100\% \right)}{k},$$ (6)

where *F* denotes the similarity between the partitioning result and the real-world observations, $N_c$ denotes the count of common nodes within each cluster formed by the clustering algorithm, $N_i$ is the count of common nodes within each cluster defined by semantic information, and *k* represents the total number of clusters.

Equation (6) computes the discrepancy in the number of common nodes between the *K* clusters generated by the clustering algorithm and the *K* clusters from manual partitioning, subsequently dividing this difference by the count of nodes in the manually partitioned cluster. These criteria can assess the deviation between the partitioning results of the clustering algorithm and the manual partitioning. They can accumulate and normalize the results obtained from each partition to more accurately represent the performance.

There are five parameters for IK-means, including *K*, *m*, *W*, *o*, and *p*. The *K* is set as 3, because there are three types of common nodes in this scenario, including conference rooms, student rooms, and staff offices. The upper limit of iteration *m* is set to 1000. The number of immediate neighboring nodes *o* is set to 5. The lower limit of the number of nodes in the reconstructed cluster *p* is set to 5.

Regarding the setting of the *W*, in order to explore the impact of *W* on the partitioning results, we conducted an experiment to determine the most suitable value for *W*. While keeping other parameters constant, we varied *W* from 0.8 to 0.45, in increments of 0.05, forming eight sets of experiments. Each set was conducted 30 times to calculate an average value. The results are presented in Table 1 and show that, when *W* is set to 0.65, the F becomes optimal. Therefore, for this experiment and this specific dataset, *W* is set at 0.65. Figure 9 shows the tendency of *F* when changing *W* by using curve fitting. The plot displays the *W* in blue and the fitted curve in red. It can be observed that the choice of *W* follows a quadratic function relationship, with the optimal value ranging between 0.60 and 0.65. The difference is not significant; thus, we choose 0.65 for this dataset.

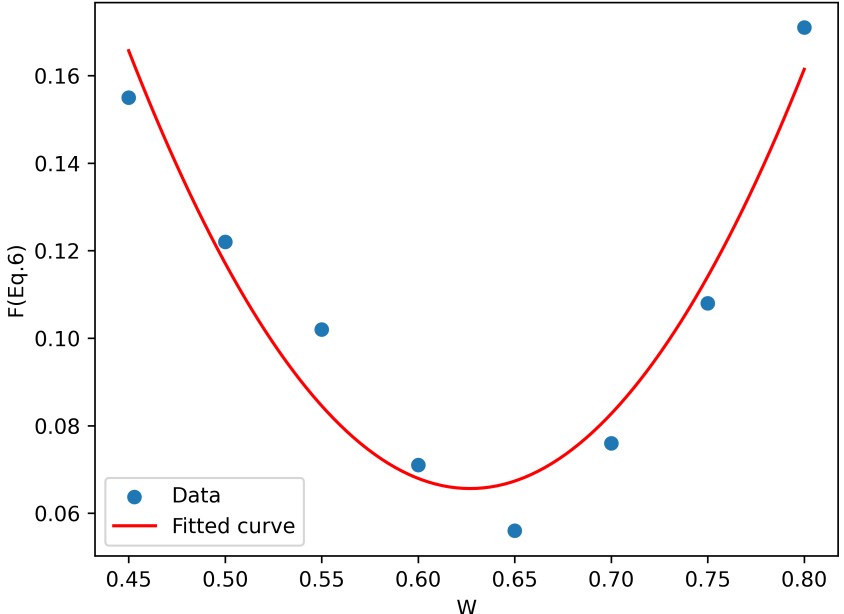

**Figure 9.** The tendency of *F* when changing *W*.

**Table 1.** Evaluation metrics of *F*.

|  | 1 | 2 | 3 | 4 | 5 | 6 | 7 | 8 |
|---|---|---|---|---|---|---|---|---|
| *W* | 0.8 | 0.75 | 0.7 | 0.65 | 0.6 | 0.55 | 0.5 | 0.45 |
| $(1 - W)$ | 0.2 | 0.25 | 0.3 | 0.35 | 0.4 | 0.45 | 0.5 | 0.55 |
| *F* (Equation (6)) | 0.171 | 0.108 | 0.076 | **0.056** | 0.071 | 0.102 | 0.122 | 0.155 |

### 4.2. Area Partition Based on IK-Means

To show what the real-world partitioning results are, we collected the actual photographs of the exits and entrances of office areas in the indoor scenario (Figure 10a), in which different shapes and colors of dots represent different clusters. Having the NRG (Figure 8b) as the input of IK-means, common nodes are initially clustered into three distinct clusters. After fixing the orphan nodes, the remaining transition nodes are further incorporated into these clusters. This process leads to the reconstruction of the three clusters, ultimately yielding the results depicted in Figure 10b. By comparing the two figures in Figure 10, we can conclude that the partitioning results of IK-means closely align with the actual partitioning in the real world.

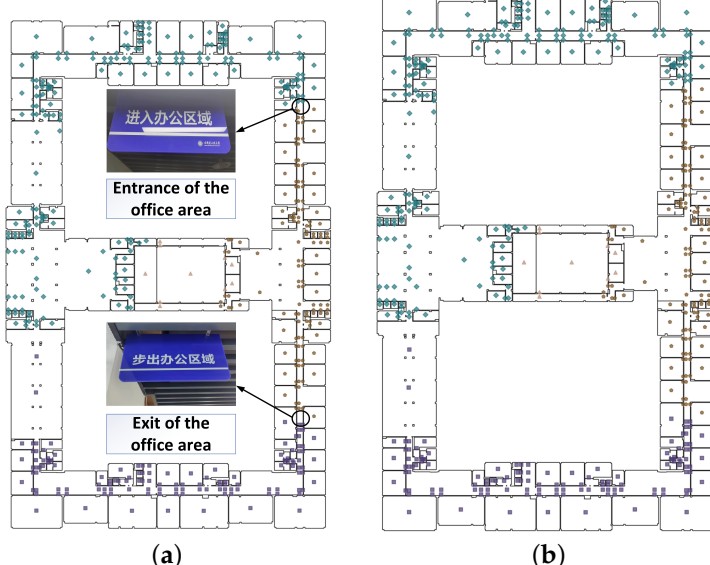

|  (**a**)  |  (**b**)  |

**Figure 10.** Partition result of IK-means and that of the real-world. (**a**) Partitions based on real-world observations. (**b**) Partitions of IK-means.

### 4.3. Results Analysis

In addition to IK-means, we take the traditional K-means (Figure 11b) and DBSCAN (Figure 11c) for comparison. IK-means is an adaptation of traditional K-means, incorporating semantic information to increase the rationality and credibility of partitioning. Improved algorithms like K-means++ do not introduce semantic information and exhibit a similar performance to that of traditional K-means in experimental testing. Moreover, DBSCAN is a density-based clustering algorithm, which has a fundamentally different clustering principle, making its comparison both logical and necessary. The partitioning results are shown in Figure 11, in which different colors in the figure represent different partitions. Figure 11a shows the partitioning result of the real world based on real-world observations. For traditional K-means, the *K* is set to 3. As for DBSCAN, considering the considerable distances between nodes in the experiment, a reasonable radius value was necessary, hence the choice of 15,000; the *eps* is set o 15,000, and the $min_{samples}$ to 5.

The K-means and DBSCAN algorithms follow physical distances for partitioning, which is reasonable to a certain degree, but deviates significantly from real-world observations. Additionally, both K-means and DBSCAN algorithms resulted in orphan nodes and disconnected clusters (as circled in the figures). In contrast, IK-means performs similarly to real-world observations, in which nodes within the IK-means partition are all interconnected by NRG. Such results indicate that the IK-means algorithm, by incorporating semantic information, achieves better partitioning effects suitable for semantically-enriched areas, compared to traditional K-means and DBSCAN algorithms.

Table 2 provides a summary of the convergence times for the three methods in eight experiments. The results show the superiority of IK-means in terms of convergence speed. Specifically, IK-means achieves the fastest processing time (only 214.015 ms), surpassing that of both K-means (average is 3480 ms), and DBSCAN (average is 1260.57 ms). The efficiency of IK-means is substantial, showcasing a noteworthy 93.85% improvement compared to K-means and an impressive 83.29% enhancement over DBSCAN. Therefore, we consider that the IK-means algorithm has a rapid convergence and the ability to yield stable results.

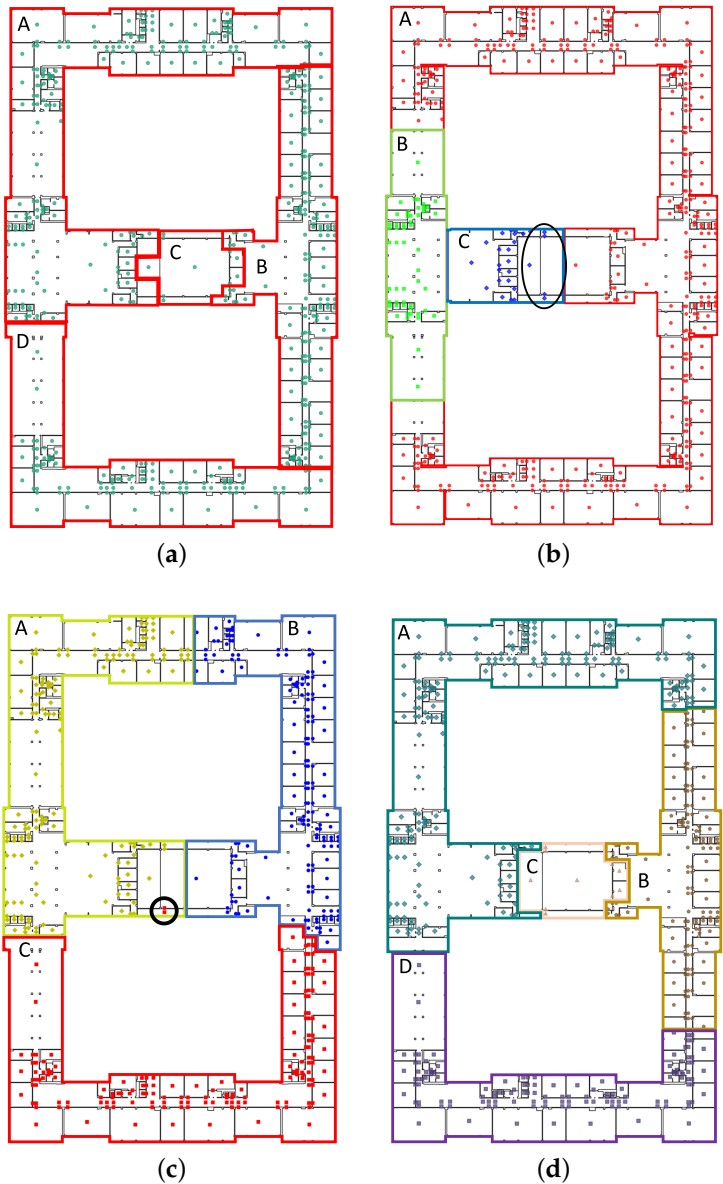

**Figure 11.** Partitioning results of real-world observations and the three algorithms. (**a**) Partitions based on real-world observations. (**b**) Traditional K-means. (**c**) DBSCAN. (**d**) IK-means. The black circle in (**b**) marks out the disconnected clusters, while that in (**c**) are the orphan nodes.

**Table 2.** Convergence time (ms).

|  | 1 | 2 | 3 | 4 | 5 | 6 | 7 | 8 | Average |
|---|---|---|---|---|---|---|---|---|---|
| K-means | 4040.29 | 3169.19 | 3203.23 | 4151.75 | 3506.62 | 3182.59 | 3329.38 | 3262.34 | 3480.67 |
| DBSCAN | 1084.18 | 1269.31 | 1280.51 | 1220.46 | 1288.78 | 1311.52 | 1364.64 | 1265.18 | 1260.57 |
| IK-means | 242.77 | 236.35 | 220.61 | 224.88 | 191.49 | 198.47 | 188.11 | 209.44 | 214.01 |

On the basis of the above results, we make an overall analysis of the partition results and efficiency of the three methods (Figure 12). It shows that, compared to K-means and DBSCAN algortihms, IK-means over-performs in terms of both partition and efficiency.

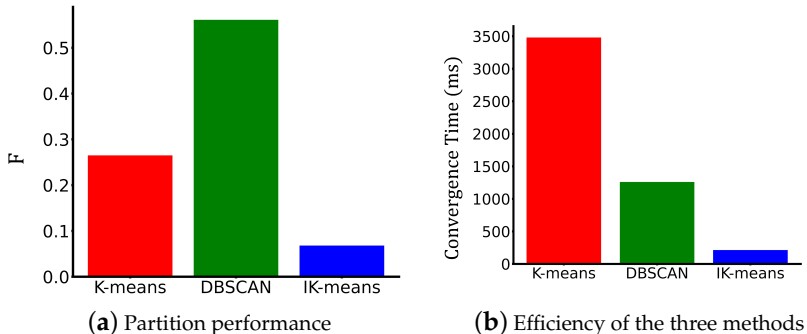

<div align="center">(<b>a</b>) Partition performance       (<b>b</b>) Efficiency of the three methods</div>

**Figure 12.** Comparison of the partition results and efficiency of the three methods.

### 4.4. Discussion

The proposed IK-means algorithm marks a significant advancement over the traditional K-means algorithm by integrating a distance measure that accounts for the semantic information of nominal attributes. This novel feature allows the algorithm to consider semantic information during partitioning, leading to enhanced node clustering evaluation and more effective partitioning outcomes. The normalization of both Euclidean and VDM distances is a key contribution because it ensures that the impact of physical and semantic information in clustering is balanced.

The parameter $W$ governs the equilibrium between Euclidean and VDM distances, allowing the algorithm to finely tune the influence of both physical and semantic distances within a specified range. This control facilitates the accurate determination of node distances. Furthermore, empirical observations suggest that the impact of $W$ selection conforms to a quadratic function distribution, showcasing optimal partitioning outcomes when $W$ is chosen from the mid-range. While preliminary experiments can be conducted for datasets of varying sizes to identify a suitable $W$, it is noteworthy that datasets of similar sizes generally exhibit a similar $W$.

The dynamic updating of clustering centers as nodes join a cluster is a critical aspect of the IK-means algorithm. This updating significantly influences the Euclidean and VDM distances when a node joins a cluster, mitigating the impact of these distances on the clustering process. By adapting to changes in the clustering structure in real time, the computation of VDM distances remains accurate, enhancing the overall partitioning effectiveness. The iteration process will be terminated when no further changes in cluster categories occur, which not only streamlines the algorithm but also improves convergence speed by reducing iterations.

The IK-means algorithm utilizes semantic information to determine an expected number of categories, effectively addressing the challenge of $K$ selection. In contrast, the traditional K-means algorithm often encounters difficulties in selecting an appropriate number of clusters ($K$). In IK-means, the initial clustering centers are informed by the semantic attributes of nodes, further reducing the influence of starting points on outcomes. However, the presence of orphan nodes and disconnected nodes in the clustering still needs to be considered further, and a better approach can be found.

The partition results of IK-means are aligned to those based on real-world observations. Traditional K-means and DBSCAN algorithms, however, fall short in addressing issues like orphan nodes and incomplete intra-cluster connectivity. The evaluation metrics presented in Table 3 show that the IK-means algorithm has a superior performance over traditional K-means and DBSCAN algorithms in terms of effectively grouping nodes with similar geometric and semantic characteristics.

**Table 3.** *F* of K-means, DBSCAN, and IK-means algorithms.

|  | 1 | 2 | 3 | 4 | 5 | 6 | 7 | 8 | Average |
|---|---|---|---|---|---|---|---|---|---|
| K-means | 0.208 | 0.304 | 0.217 | 0.225 | 0.473 | 0.284 | 0.209 | 0.203 | 0.265 |
| DBSCAN | 0.576 | 0.674 | 0.478 | 0.488 | 0.525 | 0.538 | 0.592 | 0.618 | 0.561 |
| IK-means | 0.044 | 0.056 | 0.072 | 0.124 | 0.032 | 0.054 | 0.073 | 0.092 | 0.068 |

## 5. Conclusions and Future Work

This paper presents a semantic-based clustering algorithm tailored for the partitioning of extensive indoor spaces, named IK-means. As the name implies, the presented approach is an improved version of the K-means algorithm. By incorporating both VDM and Euclidean distances to calculate the separation between the mean vector and samples, IK-means continually updates the mean vector in each iteration, enhancing convergence speed. The empirical results from using an indoor scenario dataset demonstrate that our proposed method surpasses traditional K-means and other clustering algorithms in terms of partition effectiveness and convergence speed.

The IK-means algorithm can subdivide indoor areas based on semantics, yielding notable advantages, including rapid convergence and robust partitioning. Nevertheless, several aspects remain worthy of further consideration: (i) during the data preparation, it is imperative to pre-define the attributes of nodes used as inputs; (ii) the parameter $W$ in IK-means is ascertained based on a series of experiments, necessitating fine-tuning of the threshold ($m$) during iterations; (iii) parameters ($n$, $o$, and $f$) should be adjusted during the re-correction process based on real data to achieve the overall performance; (iv) furthermore, the experiments were exclusively conducted on a single floor, which overlooks the intricate nature of indoor, three-dimensional, multi-floor environments.

Our future research will focus on adapting the IK-means algorithm for dynamic environments and implementing intelligent optimization methods to dynamically determine the hyper-parameter $W$. Ongoing efforts aim to enhance the adaptability of IK-means in 3D indoor spaces and multiple floors. Additionally, we plan to integrate IK-means with other localization and navigation techniques for increased versatility.

**Author Contributions:** Conceptualization, Kegong Shi and Jinjin Yan; methodology, Kegong Shi; software, Kegong Shi; validation, Kegong Shi, and Jinjin Yan; formal analysis, Jinjin Yan; investigation, Kegong Shi, and Jinjin Yan; resources, Kegong Shi; data curation, Kegong Shi, and Jinjin Yan; writing—original draft preparation, Kegong Shi and Jinquan Yang; writing—review and editing, Jinjin Yan; visualization, Kegong Shi and Jinquan Yang; supervision, Jinjin Yan; project administration, Jinjin Yan; funding acquisition, Jinjin Yan. All authors have read and agreed to the published version of the manuscript.

**Funding:** This research was funded by Shandong Province Natural Science Foundation Youth Branch grant number ZR2022QD121.

**Data Availability Statement:** The authors do not have permission to share data.

**Conflicts of Interest:** The authors declare no conflict of interest.

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
