# Peer review of "A Semantic Partition Algorithm Based on Improved K-Means Clustering for Large-Scale Indoor Areas"

_ijgi, doi:10.3390/ijgi13020041_

Round 1
Reviewer 1 Report
Comments and Suggestions for Authors
I think the subject of the research is very interesting and fits the scope of the journal. The methodology is adequate and the case study and discussion are added values. I find the idea very valuable and I believe the paper can be published after minor revision.
I have the following comments:
P.1, line39: ‘While this algorithm’ is ambiguous in this sentence. Clarify whether ‘this algorithm’ is the method proposed in this article or the algorithm mentioned earlier.
P.2, line 45: Is ‘simplicity and ease of implementation’ the reason for choosing k-means? I think other aspects should be emphasized to fully demonstrate the advantages of choosing k-means for improvement in this article.
P.3 line 51: This sentence is too long and not easy to understand. I suggest dividing it into several sentences.
P.4 line 72,78,146: ‘approaches/algorithms’ appears multiple times in the paper, it is recommended to use only one word for description. Check for many more sentences throughout the manuscript.
P.5 line 79,86: ‘Hierarchical clustering algorithm [12], Spectral clustering algorithm [13].’ Pay attention to whether capitalization is correct.
P.6 In the section 3.1: What is the difference between ‘points’ and ‘nodes’? Suggest the author to unify terminology to make readers clearer. Here, you can refer to the content about NRG in the Indoor GML standard for a specific word.
P.7 In the section 4.1: The content of this section can be appropriately expanded. For example, Figure 9 (b) can provide a more detailed description in the paper.
P.8 In Figure 10(c): The marked circle in the Figure 10(c) is not obvious.
Comments on the Quality of English Languageno
Reviewer 2 Report
Comments and Suggestions for Authors
In the paper, the authors propose IK-means, a modification of the K-means clustering algorithm dedicated for semantic partitioning of indoor areas.
For the purpose of semantic partitioning, indoor spaces are described via a node-relationship graph (NRG) consisting of common nodes (which represent spaces such as rooms) and transition nodes (which represent room connections such as doors and corridors).
The main difference from standard k-means clustering (aside from differentiation of nodes into common and transition ones) appears to be the introduction of distance from disordered attributes (Dvdm), which is added to the standard euclidean distance via weighted averaging. For this purpose the algorithm uses the distance ratio weighting parameter W, which (as stated in section 3.3 and shown in section 4.1) must be established experimentally, which is not optimal. As shown in Table 1, the clustering quality can vary in a sinusoidal manner for linearly decreasing values of W (in particular, W equal 0.5 and 0.6 both give better results than W equal 0.55), which may make finding the best solution a tedious process. Moreover, in the later sections of the algorithm description (particularly in the first paragraphs of section 3.5) the consecutive steps of the algorithm are described very briefly, which negatively impacts comprehensiveness. I would advise a careful correction of this segment of the paper (please see detailed recommendations below). I would also recommend adding intermediate figures to visualize eg. clustering of transition nodes before the subsequent cluster reorganization.
I also have issues with the presentation of results (section 4). Only one example dataset is processed, and despite the fact that it is a moderately large graph, it is not enough to test a new methodology. At least three examples should be given: a small graph, a medium sized one and a large one. This setup would enable the comparison of algorithm effectiveness for different input data sizes. Using more examples would also serve as a basis for an in-depth discussion of the methods for discovering the optimal value of the weight parameter W, which is a necessary addition in my opinion. Moreover, the chosen input parameters of DBSCAN should be listed in section 4 alongside that of K-means. Without this information it is difficult to assess the presented performance results.
In addition, the discussion section should be rewritten to present a critical assessment of the achieved results (currently it does not).
Until the aforementioned issues have been resolved, I cannot recommend the paper for publication.
Detailed recommendations:
Line 80: This sentence needs to be rewritten. Did you mean to begin with "K-means is a ..."?
Lines 165-166: This sentence is unnecessarily complex and should be rewritten.
Line 316: Did you mean: "first it is necessary to compute"?
Line 317: Did you mean "Next, it is necessary to determine"?
Line 327: Did you mean "If the node count (...) should fall"?
Line 352: Did you mean "Each node has three attributes"?
Figure 6: What do you mean by "actual example"? Also, the first word of the sentence should be capitalized.
Figures 8-10: Please add a legend describing what is represented by each colour.
Language quality is generally fine, however I have spotted some grammatical errors (see detailed recommendations).
Reviewer 3 Report
Comments and Suggestions for Authors
Authors propose improved K-means clustering method to enable spatial data (specially focused on indoors space with multiple rooms interconnected with doors and corridors) to be organized into clusters. Improvement is supported with taking into account semantic data that describe each spatial unit.
Authors should consider some manuscript improvements:
1. Common nodes in NRG are related to spaces with particular use, such as rooms, elevators, stairs...It is not precise to say that common nodes are characterized by "semantic information" - see line 185. It has been somewhat explained with examples of relevant semantic attributes presenting their purpose or structure - of shopping malls and hospitals in lines 235-237, but still not precise enough. There should be more precise set of relevant attributes for spatial semantics related to particular types of utilized spaces, such as rooms in hospitals, shopping malls, schools etc.
2. Formation process of clustering is not focused enough to the semantics alignment of nodes. Initially, having the contribution aim of this paper to emphasize semantics, it is stated that : A) "The formation of clusters K is contingent upon the primary semantic features of these common nodes. These primary semantic features represent the classifications of semantic information for these nodes and serve as the basis for partitioning." (lines 250-252), B)"the common nodes displaying K distinct major types of features, we arbitrarily select one common node to serve as the initial clustering center" (lines 254, 255). Later, Euclidean distance is introduced, based on physical distance between nodes.
3. What is the meaning of disordered attributes? (line 261) Does this include the existing of the opposite term - ordered attributes? What is the criteria for ordering attributes? Are they ordered within one particular case (i.e. node) or there is ordering between attributes values within one cluster, having multiple nodes included and related to the cluster center?
4. Each abbreviation should be explained with full names at the first appearance in text. For example, there is no explanation for HVDM (firstly mentioned in line 270).
5. Selecting of cluster center is not precise - In line 230 it has been said "classify nodes with similar semantic information into the appropriate clusters.", so this means that cluster is a set (partition) of nodes having similar semantic information. Prior to classification of nodes into some semantic-based categories, these categories (as initial groups of nodes, i.e. initial clusters) are determined according to values in semantic information attributes, as stated in 223-224 lines "prior to the clustering process, we can determine the number of partitions, denoted as K, categorize samples into k groups based on their semantic information". In line 225-226 it is stated that "one node is randomly selected from each of these initially classified sample categories to serve as an initial clustering center".
6. Figure 3. is not aligned with the text describing it. In lines 294-295 there are 3 clusters mentioned that have 3 colors for nodes (red, green, blue) visible at Figure 3, while this figure does not contain 3 colors for nodes, but only green for lines between nodes and blue for nodes.
7. Algorithm of IK-means clustering represents the essence of this manuscript contribution. Therefore, it should be clearly presented. Currently, it is not clear what is the meaning of all mathematical symbols used in the algorithm, so they should be explained next to the proposed equations, such as "Sample set D = {x1, x2, x3, · · · , xm}, number of clusters K, threshold m" should be enhanced with explanation what are xi.
8. Explanation of orphan node is not clear enough. For example "An orphan node belong to a different cluster class than its nearest geometrically associated neighboring node, while it is geometrically distant from the node in its own cluster class. This can be explained by that clustering primarily considers the distance between nodes and the cluster center without taking into account the connectivity among nodes, orphan nodes frequently emerge during clustering." (lines 289-293). These sentences are not precise, since the word "distance between nodes" could be understood as physical distance, not "semantical distance". It is necessary to clearly point out the prioritization of grouping a node into a cluster - is it semantical closeness with values of relevant attributes with first priority, while physical closeness is taken secondly as a criteria.
9. In this manuscript it has been stated that fixing orphan nodes is performed based on physical closeness and the semantic attributes of this node are "put aside", having the node primarly categorized into a cluster based on semantics now makes him shifted into another cluster which is phyisically closer. It has been stated in lines 306-307 "If the number of neighboring nodes belonging to different cluster classes surpasses the number of clusters to which the node belongs, then the node’s cluster class is updated to match the cluster class hosting the maximum number of neighbor nodes." This way, physical closeness to a group of surrounding nodes becomes more relevant then semantics. This is opposite to the contribution that was aim in this paper, i.e. stated in lines 404-405: "This feature allows the algorithm to factor in semantic information during partitioning, thereby enhancing node clustering evaluation and yielding superior partitioning outcomes." If physical closeness become criteria for node fixing, then it becomes more important than semantic criteria for having a node classified into some category i.e. partition/cluster. And this emphasis of physical closeness in node fixing as a rule is quite opposite to the conclusion "This paper presents IK-means, a semantic-based clustering algorithm tailored for the partitioning of extensive indoor spaces." (line 436).
10. Case study is provided in section 4. Still, there are no clear set of attributes that could be used for making partitions, i.e. clusters. There is only said that there are different categories of nodes, according to their purpose. In lines 352-355 there is said: "Each nodes has three attributes: {ID, Type, Attribute}, “ID" records the id of each nodes, "type" keeps the type of nodes and determines whether the node participates in IK-means (e.g., “Common nodes", “Transition nodes"), “attributes" records the specific semantic representations of the nodes (e.g., “conference rooms", “stuff offices", “door nodes", “corridor nodes"). So, under semantics, there is actually a categorization of room i.e. node purpose, not real attributes and values. Therefore, how could it be computed wheather a node is close or distant to the purpose of the room as the only room/node attribute?
11. Table 1 is not precise regarding criteria for classification. There are 8 categories i.e. 8 clusters, but it is not clear what is the attribute having these categories possible and what are their values, making the basis for classification. There are only numbers 1-8 as table columns. The same issue continues to be present in Table 2, Table 3...
12. Table 1. does not provide explanation for W, 1-W and F. It is usual to have these symbols explained below the table or within the table caption.
Comments on the Quality of English Language
Regarding English writing, sometimes there are sentences that are not understandable enough. Example of such sentence is: "sample is randomly selected as the initial clustering center {μ1, μ1, · · · , μ1} from each of the corresponding attribute categories" (in Algorithm of IK-means clustering, starting at line 283). Firstly - sample is the collection of all nodes, so sample is used to be partitioned. One of nodes from the sample could be used as clustering center, but prior to this partition and clustering center selection, there should be selection of relevant attributes and determination of values related to each node. So, sentences in the algorithm should be more precise and more understandable.
Reviewer 4 Report
Comments and Suggestions for Authors
The paper is about clustering indoor spaces based on their semantic information. It proposes a new approach based on k-means. The paper discusses problems with current methods and aims to solve them to make accuracy and efficiency better. Finally the model provides some modifications on limitations of k-means which is called IK-means.
Include some quantitative results or specific metrics to highlight the improvement over existing methods.
Improve the readability:
· Patients can better estimate the areas they need to visit, such as 32 inpatient wards and diagnostic areas, reducing waiting times and confusion and enhancing 33 the overall patient experience.
· Among various approaches/algorithms for partition, K-means is one of the commonly used machine learning algorithms for partitioning n observations into k clusters, where each observation is assigned to the cluster with the nearest mean, which serves as a prototype for that cluster.
· Considering that K-means has the advantages of being simple and easy to implement, this paper proposes an improved K-means clustering algorithm (IK-means), which aims to partition large-scale indoor areas based on semantics to improve the accuracy and efficiency of large-scale indoor space partitioning.
· In contrast to traditional K-means and DBSCAN, our approach leverages semantic information associated with known indoor room node attributes, resulting in partitioning results with lower evaluation functions.
The author claims that K-means is widely used as a machine learning algorithm for partitioning; however, not enough/recent references are provided.
The paper doesn't discuss about other existing clustering methods or why it focus on k-means, specially in introduction. It should mention more advanced methods to compare with, not just the two. Or, at least it should explain why it chose these two methods in particular for comparison.
Related work:
The references in the related work section only cover older years (the newest is for 2019), and they don't include information about recent studies in regional clustering. Additionally, there is no mention of the latest works that may have addressed the limitations of existing methods in recent years.
The explanation for choosing K-means for improvement among the mentioned algorithms could include discussing its potential in semantic partitioning and leveraging information from previous studies in this area and discuss about the semantic information modeling.
Method:
Improve readability:
· This algorithm leverages the VDM to measure the distance between unordered attributes and subsequently normalizes this distance following its amalgamation with the Euclidean distance.
· This can be explained by that clustering primarily considers the distance between nodes and the cluster center without taking into account the connectivity among nodes, orphan nodes frequently emerge during clustering.
· If the number of neighboring nodes belonging to different cluster classes surpasses the number of clusters to which the node belongs, then the node’s cluster class is updated to match the cluster class hosting the maximum number of neighbor nodes.
It relies on a method called VDM to calculate distance, so it's important to explain exactly what this distance means and how it impacts grouping things based on their semantic.
The paper discusses improving the traditional k-means approach, so it's better to have preliminary information about the approach.
The paper discusses finding the optimal number of clusters (k) in a dataset. It considers the semantic information of the data points in the clusters and claims to apply this to the traditional k-means method. However, it doesn't look like a novelty; instead, it appears to be a case study where they use k-means with additional details about the data context.
There is a misspelling in Algorithm 1 regarding the initial clustering centers.
The example of orphan node could be explained with more details. It's not clear or there is a problem with figure 3.
Implementation
There is not any explanation for table 1.
It could use different state of the art and the newest methods to compare the proposed method.
It only looked at one measure to compare the models, and there was a big improvement. Maybe it should also check the models using other measures.
Comments on the Quality of English Languagemainly consider readability and avoid passive voice sentences.
Round 2
Reviewer 2 Report
Comments and Suggestions for Authors
The updated paper has addressed the majority of my concerns. This being said, I believe the discussion section needs to include elaboration on how establishing the ratio weighting parameter W impacts the performance of the proposed algorithm, and whether the performance profile demonstrated in the previous section remains true for graphs of different sizes.
Detailed recommendations:
Lines 46-48: This sentence should be rewritten to make it clearer.
Aside from the mentioned sentence (Lines 46-48) I see no issues with language quality.
Reviewer 3 Report
Comments and Suggestions for Authors
The corrected version of submitted manuscript has appropriate scientific structure and acceptable level of English writing. Sentences are more understandable as being split and well organized.
However, still there are some aspects that should be considered for refining:
1. The term - semantics attribute is used as synonym for the usage type of a space unit (room). In lines 196-198. it is said "Depending on the specific usage space, they possess various related attributes." After this sentence, next sentence explains the case of hospitals, but under attributes, it lists specific types of spaces by their utilization, not attributes.
2. This paper proposes DVDM as a distance metrics related to nominal (attributes i.e. space utilization type) data. This metrics is used for calculating semantic distance of a node to the cluster center. Equation (1) in line between 280 and 281 mentions xil and wil as "unknown" (which is not acceptable - to use "unknown" in the formula).
3. Under the formula (1) xil is noted as the lth attribute of instance xil - this is not acceptable to have the same annotation for an attribute and node, potentially being part of a cluster.
3. Within the formula (1) it is said that there are multiple attributes related to a single node (space), but it is not the case in this approach - in this paper there is only space utilization type considered as an attribute...
4. Class and cluster are equally used. This should be uniformed across the whole manuscript (to use only cluster as a term or to explain the difference, if exists).
5. What is the meaning of "semantic distance"..? It seems that there are no semantic similarities, synonyms and other situations considered, but only "class c appears within the attribute value"...this tells that only equal names of space types are considered in nodes clustering (equality of strings, in fact)...So here we can not speak about "semantic distance" but only about textual equality of cluster name to utility type of space (node) to be included in the cluster. So it seems that the criteria for clustering is boolean (node has or has not the same utility type as all nodes in the cluster). How can we speak about "distance" at all?
6. Authors should consider having more precise terminology. In this sentence "Sample set D represents the collection of nodes involved in clustering, and xi denotes each individual cluster sample, specifically referring to nodes participating in the clustering process, and m represents the total number of samples participating in clustering." - node is considered equal as sample...Usually, sample is related to a collection of items, not particular item = node. So it would be better to use "node example", not "sample" as a term. So, D is a collection of nodes or node examples.
